# Freshwater Mussels Show Elevated Viral Richness and Intensity during a Mortality Event

**DOI:** 10.3390/v14122603

**Published:** 2022-11-23

**Authors:** Jordan C. Richard, Eric M. Leis, Christopher D. Dunn, Cleyo Harris, Rose E. Agbalog, Lewis J. Campbell, Susan Knowles, Diane L. Waller, Joel G. Putnam, Tony L. Goldberg

**Affiliations:** 1Department of Pathobiological Sciences and Freshwater & Marine Sciences Program, University of Wisconsin-Madison, Madison, WI 53711, USA; 2Southwestern Virginia Field Office, U.S. Fish and Wildlife Service, Abingdon, VA 24210, USA; 3La Crosse Fish Health Center, Midwest Fisheries Center, U.S. Fish and Wildlife Service, Onalaska, WI 54650, USA; 4Michigan Department of Natural Resources, Waterford, MI 48327, USA; 5U.S. Geological Survey, National Wildlife Health Center, Madison, WI 53711, USA; 6U.S. Geological Survey, Upper Midwest Environmental Sciences Center, La Crosse, WI 54603, USA

**Keywords:** bivalve, virome, freshwater mussel, mass mortality, die-off, unionid, virology, invertebrate, biodiversity

## Abstract

Freshwater mussels (Unionida) are among the world’s most imperiled taxa, but the relationship between freshwater mussel mortality events and infectious disease is largely unstudied. We surveyed viromes of a widespread and abundant species (mucket, *Actinonaias ligamentina*; syn: *Ortmanniana ligamentina*) experiencing a mortality event of unknown etiology in the Huron River, Michigan, in 2019–2020 and compared them to viromes from mucket in a healthy population in the St. Croix River, Wisconsin and a population from the Clinch River, Virginia and Tennessee, where a mortality event was affecting the congeneric pheasantshell (*Actinonaias pectorosa*; syn: *Ortmanniana pectorosa*) population. We identified 38 viruses, most of which were associated with mussels collected during the Huron River mortality event. Viral richness and cumulative viral read depths were significantly higher in moribund mussels from the Huron River than in healthy controls from each of the three populations. Our results demonstrate significant increases in the number and intensity of viral infections for freshwater mussels experiencing mortality events, whereas individuals from healthy populations have a substantially reduced virome comprising a limited number of species at low viral read depths.

## 1. Introduction

Freshwater mussels of the family Unionidae are among the most imperiled fauna on earth. Ten percent of the ~300 species of North American unionids are already extinct [1], while approximately two-thirds are threatened, endangered, or vulnerable [2,3]. Freshwater mussels contribute valuable ecosystem services, including increased water clarity via filter-feeding and removal of suspended particulates [4], food web enhancement [5], and increased physical habitat complexity [6]. Habitat destruction, effects from invasive species (e.g., the zebra mussel *Dreissena polymorpha*, quagga mussel *D. bugensis*, and the Asian clam *Corbicula fluminea*) [7], commercial over-harvest [8], and pollution are some of the factors implicated in freshwater mussel population declines [9]. However, many unionids have suffered enigmatic mass mortality events (MMEs) and rapid population losses that have occurred without obvious causes [10,11]. Restoration strategies for freshwater mussel conservation largely rely upon captive propagation of imperiled species to augment wild populations and translocation from healthy populations to reestablish extirpated populations [12]. These conservation efforts necessitate a better understanding of the relationship between infectious disease, MMEs, and population extirpations, as restoration efforts could be counterproductive if they inadvertently introduce novel pathogens when introducing or translocating mussels.

We previously found an association between a novel densovirus (Clinch densovirus 1) and mass mortality of pheasantshell mussels (*Actinonaias pectorosa*; syn: *Ortmanniana pectorosa*) in the Clinch River, Virginia and Tennessee, USA [13]. In general, however, associations between viral disease and mortality events in freshwater mussels are understudied, as cell culture isolation of mussel viruses has largely been limited to those that can replicate in fish cells [14]. The only viral disease in the Unionidae for which detailed pathogenic mechanisms are known is Hyriopsis cumingii plague virus, a member of the *Aerenaviridae* that causes mass mortality of *Hyriopsis cumingii* in aquaculture settings [15]. Other viruses described for bivalve mollusks are primarily in marine species of economic and agricultural importance, such as Ostreid herpesvirus-1 (OsHV-1) in oysters (*Crassostrea gigas*). High-throughput transcriptomic data have recently led to the characterization of a large number of viruses in both marine and freshwater mussels, but the relationship between these viruses and mussel health is unclear [16].

The objective of this study was to investigate viruses associated with an unexplained mortality event of mucket (*Actinonaias ligamentina*; syn: *Ortmanniana ligamentina*) in the Huron River, Michigan, in 2018 and to compare the mucket virome between mortality sites and unaffected populations. We sought to identify viruses in affected Huron River mucket that could be potential causes of the observed mortality. We also examined patterns of virus richness and viral read depths between the affected Huron River mucket population and that of healthy mussel populations. Our findings reveal marked differences in mussel viromes between affected and unaffected populations and shed new light on the role of viruses in unionid MMEs.

## 2. Materials and Methods

### 2.1. Experimental Design and Field Sampling

We sampled mucket from three rivers (Figure 1, Table 1). In September 2019, we responded to a report of unionid mortality on the Huron River, Michigan, where we sampled 8 mucket (5 moribund and 3 apparently healthy). Based on observations of recurring annual mortality in other unionid populations, we revisited the Huron River site one year later in September 2020. During the revisit, we observed no evidence of morbidity or mortality, and we sampled 4 apparently healthy mucket for comparison to those collected during the previous mortality event. We sampled 8 healthy mucket in August 2018 from a reference (control) population in the St. Croix River, Wisconsin, where there was neither observed mortality nor apparent water quality issues. In the Clinch River, Virginia and Tennessee, we sampled 9 healthy mucket in August, September, and October 2018 during a survey of a MME (described in [13]) affecting the congeneric pheasantshell at 4 sites. During the Clinch River MME, we saw no evidence of elevated morbidity or mortality in mucket, which were all considered apparently healthy despite their proximity to dead and dying mussels of other species. We sampled mucket non-lethally by collecting hemolymph via syringe from the anterior adductor muscle sinus, as previously described in [13]. Samples were stored in microcentrifuge vials, placed immediately on dry ice in the field, and transported to −80 °C freezers where they were stored until laboratory processing. Complete details for each individual sampled are in Appendix A.

### 2.2. Viral Nucleic Acid Extraction and Sequencing

We processed mussel hemolymph samples for virus characterization using previously described methods [17]. Briefly, we centrifuged hemolymph at 10,000× *g* for 10 min, transferred the supernatant to clean vials, and concentrated virus particles by centrifugation at 25,000× *g* for 3 h. We extracted total nucleic acids with the QIAamp MinElute Virus Spin Kit (Qiagen, Hilden, Germany), converted RNA to double-stranded cDNA using the Superscript IV system (Thermo Fisher, Waltham, MA, USA) with random hexamers and NEBNext Ultra II Non-Direction RNA Second Strand Synthesis Module, and prepared DNA libraries using the Nextera XT DNA Library Preparation Kit (Illumina, San Diego, CA, USA). We sequenced libraries on a MiSeq instrument (V2 chemistry, 300 cycle kit; Illumina, San Diego, CA, USA). Samples from each population (Clinch River, St. Croix River, and Huron River) were sequenced on separate runs, and total read depth of each run was assessed to ensure that each run was of approximately equal depth.

### 2.3. Sequence Analysis and Phylogenetics

We used CLC Genomics Workbench version 20.1 (Qiagen, Hilden, Germany) to quality-trim demultiplexed reads to ≥Q30 and discarded short reads (length < 50 nt). We then filtered reads to mask low-complexity regions, laboratory contaminants, and eukaryotic reads using an in-house database. We assembled reads into contiguous sequences (contigs) for each individual using metaSPAdes v3.15.2 [18]. We queried assembled contigs >500 nt length against sequences of viruses in GenBank at both the nucleotide (BLASTn) and deduced amino acid (BLASTx) sequence levels. We retained all putative eukaryotic viral contigs matching virus sequences with E-values < 10^−20^ that did not have equivalent or better matches to bacterial, bacteriophage, or eukaryotic sequences. To detect viruses with circular genomes, we checked contigs for identical k-mers at both ends. When such k-mers were detected, we trimmed the repeat sequence for one end, circularized the sequence in CLC genomics workbench, and selected a point that did not interrupt any open reading frames as the sequence origin. We removed putative viral contigs matching known laboratory contaminants [19,20]. For identical viruses present in more than one individual (determined as contigs >97% sequence identity for the full length of the contig), we compared contigs among individuals and selected the individual with the greatest number of reads and the longest contig as the representative genome for each virus.

For all identified viruses, we inferred phylogenetic trees based on nucleotide sequence alignments of the replication-associated gene sequence (when available) to related sequences in GenBank. We used TranslatorX [21] to apply a codon-based version of the Prank algorithm [22] and Gblocks [23] to remove poorly aligned regions. We used PhyML 3.1 [24] with smart model selection to infer phylogenies and ran 1000 bootstrap replicates to evaluate statistical confidence in clades. We then visualized resulting phylogenetic trees using FigTree v1.4.4. To further characterize viral genomes, we used Cenote-Taker 2 in annotation mode [25] and Conserved Domain Database [26] matches from BLASTx searches to label open reading frames with putative functional annotations (Appendix A). Because many identified viruses are most similar to unclassified viruses, we assigned virus names using names derived from the sampling site, genus, and species of the samples (e.g., flactilig virus 1 = Flatrock Michigan, *Actinonaias ligamentina*) and we did not include taxonomic descriptors in virus names, so as to avoid confusion from future taxonomic revisions.

### 2.4. Statistical Analysis

To avoid biases associated with variable sequence depth between individuals, we randomly sampled 1,000,000 reads per individual from libraries after quality trimming and discarding short (<50 nt) reads. For each individual, we mapped sampled reads to mussel virus contigs generated at a stringency of 95%. We counted a virus as present in a sample if it contained reads that mapped to a contig with ≥95% similarity over the full length of the read. We then calculated a measure of viral concentration normalized by contig length to account for differing target sequence lengths for each virus [27]. The resulting measure, viral reads per million per kilobase of target sequence (vRPM/kb), has been validated by comparison to quantitative real time polymerase chain reaction [17]. We also calculated a related measure of cumulative viral concentration for all viruses in each individual (hereafter, “viral intensity”) by summing all reads mapped to all virus contigs and normalizing by the cumulative length of all summed target sequences (in kb).

We compared measures of viral richness and cumulative viral intensity for each population using ANOVA with Tukey HSD multiple comparisons with significance thresholds of *p* < 0.05. For samples from the Huron River mortality event, we compared virus prevalence and concentration for all samples together, as well as for the subgroups of cases versus controls, and 2019 (pooled cases and controls) and 2020 (controls only). In total, we modeled six one-way ANOVAs with Tukey HSD to examine viral richness and cumulative viral intensity. Input values for the Clinch River and St. Croix River did not differ between each test, whereas the values for the Huron River iteratively involved: (1) all samples (comparison of 3 groups total), (2) independent groups for Huron River cases and controls (comparison of 4 groups total), and (3) Huron River groups stratified by year and clinical status (2019 cases, 2019 controls, 2020 controls, comparison of 5 groups total). To examine associations between individual viruses and disease in the Huron River, we used t-tests to compare average viral read depths for each virus (as vRPM/kb) between cases and controls. To rank the importance of individual viruses in distinguishing cases from controls in the Huron River, we used the randomForest package [28] in R version 4.1.2 [29] to model the relationship between clinical status and viral read depths of each virus. We computed the model using the function *randomForest*, with settings ntree = 50,000 and mtry = 3, followed by the importance function to rank the relative contribution of each virus in distinguishing between cases and controls via the mean decrease in Gini values. We also calculated Pearson correlation coefficients (*r*) and their statistical significance for all pairwise read depth combinations in mussel samples coinfected with two viruses using Hmisc [30] in R, limiting the analysis to comparisons for which ≥4 samples were coinfected to avoid spurious results.

## 3. Results

### 3.1. Virus Characterization

Total read depths from all individuals sequenced for each population were as follows: 28,215,178 (Clinch River), 31,560,512 (St. Croix River), and 27,337,570 (Huron River). After quality trimming, average sequence depth was 2,532,757 (standard deviation (SD) 777,402) reads per individual with an average length of 122.3 nt (SD = 10.4 nt). From these data, we identified 38 viruses, most of which likely represent novel taxa, from contigs ranging 745–12,425 nt (Appendix A). Most viruses were only distantly related to known invertebrate viruses based on sequence similarity. We identified contigs representing 7 viruses from the Clinch River, 6 from the St. Croix River, and 26 from the Huron River. One virus (flactilig virus 1) was detected independently in both the St. Croix and Huron River samples with >97% sequence similarity between contigs from the two samples. Viruses identified in apparently healthy mussels included picorna-like viruses, dicistroviruses, tombusviruses, picobirnaviruses, and nodaviruses. Viruses identified in samples collected during the Huron River mortality event included picorna-like viruses (including dicistroviruses), tombusviruses, nodaviruses, CRESS viruses, and single representatives each of a narnavirus, densovirus, calicivirus, and reovirus. Maximum likelihood phylogenetic trees for all viruses are available in Appendix A. In the Huron River virome during the mortality event, the picornavirus and dicistrovirus members accounted for more than half of the viruses present, while other groups were represented by fewer members (Appendix A).

### 3.2. Virus Richness, Read Depths, and Intensity Statistics

Mussels from the unaffected St. Croix River had the lowest viral richness (mean = 4.25) and intensity (mean = 0.34 vRPM/kb) (Figure 2). Mussels from the Clinch River had intermediate values for both measures, with 5.33 viruses on average, and a mean viral intensity of 0.73 vRPM/kb. Collectively, the Huron River mussels had the highest average number of viruses per individual (mean = 11.92) and viral intensity (mean = 1.18 vRPM/kb). This pattern was stronger when samples were stratified by clinical status and collection date. During the mortality event in 2019, Huron River mussels (*n* = 8; 5 cases and 3 controls) averaged 15.75 viruses per mussel with a viral intensity of 1.65 vRPM/kb compared to 4.25 viruses per mussel with a viral intensity of 0.25 vRPM/kb in 2020 (*n* = 4; all controls). Moribund case mussels from the Huron River (*n* = 5; 2019 only) averaged 16.60 viruses per mussel with an average viral intensity of 1.63, compared to apparently healthy control mussels (*n* = 7; 2019 and 2020), with 8.57 viruses per mussel and viral intensity of 0.86 vRPM/kb on average. Cases and controls from the Huron River for 2019 alone (during the mortality event) had approximately equal values for average prevalence (2019 controls = 14.33, 2019 cases = 16.60) and average viral intensity (2019 controls = 1.68 vRPM/kb, 2019 cases = 1.63 vRPM/kb). Average values for all groups and subgroups are in Appendix A.

Random forest analysis indicated that the three DNA viruses with circular genomes detected from the Huron River samples (flactolig virus 1, flactolig virus 2, and flactolig virus 3) were the most important in distinguishing between cases and controls. Visualization of the mean decrease in Gini demonstrated that the viruses with circular DNA genomes each had a substantially larger effect on distinguishing cases from controls compared to all other viruses (Figure 3). The next most important viruses were flactilig virus 16 and flactilig virus 21 with they and the remaining 21 viruses showing a relatively even decline in importance (Figure 3).

Seven individual viruses had statistically higher viral read depths in Huron River cases versus pooled controls (i.e., controls from 2019 and 2020 combined), including two dicistroviruses, a narna-like virus, a densovirus, and all three viruses with circular DNA genomes identified from the Huron River population. We further examined the viral read depths of controls stratifying by year. The three viruses with circular DNA genomes were the only viruses with significantly higher viral read depths in cases when compared separately to both unpooled control groups (2019 and 2020). These viruses (flactolig virus 1, flactolig virus 2, and flactolig virus 3) had viral mean loads of 2.37, 1.72, and 1.19 vRPM/kb in cases versus 0.37, 0.16, and 0.17 vRPM/kb in controls (both years pooled), respectively. Flactilig virus 4 and flactilig virus 8 (dicistroviruses) had significantly higher viral read depths in cases than in pooled controls, and flactilig virus 8 was observed only in cases (present in 3/5 cases and 0/7 controls). Flactilig virus 21 (densovirus) had signficantly higher viral read depths in cases than in pooled controls, and was present in 4/5 cases compared to 1/7 controls. Flactilig virus 22 (narna-like virus) also had statistically higher viral read depths in cases versus pooled controls, and was present in 4/5 cases versus 2/7 controls.

The populations in the Clinch River, St. Croix River, and 2020 Huron River control subgroup (apparently healthy subgroup) did not differ significantly from each other for either viral richness or intensity (Figure 2). In contrast, the subgroups of 2019 Huron River controls and 2019 Huron River cases were significantly higher than both the St. Croix and Clinch River population values for both the viral richness and intensity measures. Huron River cases also had significantly higher viral richness and intensity measures than pooled 2019 & 2020 Huron River controls.

Viromes from 2019 cases and 2019 controls in the Huron River had similar composition in terms of both individual virus prevalence and cumulative viral richness (Figure 4). Of the 26 viruses detected from the Huron River population, 23 viruses had higher average viral concentrations in 2019 cases than in 2019 controls or in combined controls from 2019 and 2020. Three viruses were detected only in cases from the Huron River (flactilig virus 8, flactilig virus 18, and flactilig virus 19). The most prominent difference between Huron River cases and controls was the presence and intensity of the three Huron River cicular viruses (flactolig virus 1, flactolig virus 2, and flactolig virus 3).

### 3.3. Comparisons of Virome Composition within and between Populations

Mussel viromes were highly population-specific. This trend was consistent over time, in that we identified no new Huron River viruses in the 2020 follow-up sampling (i.e., all viruses from 2020 samples were previously detected in multiple individuals from 2019 samples). All three populations contained multiple highly prevalent novel small DNA viruses with circular genomes (Clinch = 4, St. Croix = 2, Huron = 3). Of 38 viruses described in this study, 32 were detected only in a single population of mucket. Flactilig virus 1 was the only virus in this study independently characterized from two populations (i.e., nearly identical contigs >1000 nt were present in the St. Croix and Huron River populations). This nodavirus was present in 63% of mussels from the St. Croix River and 83% of mussels from the Huron River, but was not detected in the Clinch River. Flactilig virus 9 was the only virus found in all 3 populations (6 individuals from the Huron River, 4 from the St. Croix River, 3 from the Clinch River), although it was present at very low intensity and genome coverage in the latter two populations. The other four viruses detected in more than one population were found only at low levels in a single individual from a second population (clictilig virus 1, clictilig virus 2, flactilig virus 5 and flactilig virus 7).

In the Clinch River, where we collected individuals from four locations throughout a 35 km stretch of river, 6/7 viruses were observed at all sites. Most Clinch River viruses were found in all 3 sampling months (August, September, and October), with the exceptions of clictilig virus 1 (detected in a single individual from October) and clictilig virus 2 (not detected in October samples). Most Clinch River mucket viruses were observed in the majority of individuals, with 5/7 viruses detected in 7 or more of the 9 individuals sampled. The other two Clinch River mucket viruses were detected in 1 and 3 individuals. In the St. Croix River, most of the 6 viruses detected were similarly common among individuals, with 4/6 viruses detected in ≥5/8 individuals. The other two viruses were detected in 1 and 2 individuals. In the Huron River, 16/26 viruses were only detected in the year of mortality (2019). Of the remaining 10 viruses detected in the Huron River, all viruses detected in the following year (2020) were also detected in the year of mortality.

### 3.4. Correlations among Read Depths of Different Viruses

Read depths between viruses in dually infected mussels showed few correlations in healthy populations but strong positive correlations in the Huron River population (Appendix A). In the Clinch River, only 2 of the 21 potential pairwise comparisons were positively correlated. Clictolig virus 1 and clictilig virus 3 were very strongly correlated (r = 0.98), while Clictolig virus 2 and clictolig virus 3 (both circular DNA viruses) were also positively correlated (r = 0.79). In the St. Croix River, only 1/15 pairwise combinations was positively correlated: scracolig virus 2 and scractlig virus 3 (r = 0.97). In the Huron River, 43/325 pairwise combinations were positively correlated, with an average r among significant correlations of 0.91. Flactilig viruses 1 and 2 (nodaviruses) were correlated only with each other (r = 0.65). With the exception of flactilig virus 8, which was not correlated with any other viruses, all dicistroviruses from the Huron River were strongly positively correlated with each other (average r = 0.96) and with the 3 circular DNA viruses (average r = 0.89). Similarly, read depths among Huron River circular viruses were all strongly correlated with each other (average r = 0.99).

## 4. Discussion

We identified 38 viruses from 29 mucket in 3 populations from different watersheds. The majority of viruses were previously uncharacterized, with most distantly related to viruses known to infect invertebrates [31,32]. Unaffected populations had 2–4 unique ssDNA viruses with circular genomes and a small number (~3) of ssRNA viruses in each population. The population experiencing a mortality event also had 3 unique ssDNA viruses with circular genomes and were infected with more viruses (*n* = 23), the majority of which were ssRNA viruses resembling members of the *Picornavirales*.

Composition of the mucket virome was population-specific: 32/38 viruses were observed only within a single population. Further, we found little similarity between viruses identified from Clinch River mucket and viruses identified in a previous study [13] of the congeneric pheasantshell, which was surveyed simultaneously. Our data therefore indicate low inter-population but high intra-population virome compositional similarity, as viromes sampled at multiple time points from a given population were highly similar, but few viruses were detected in multiple populations. Such species- and population-specificity is noteworthy for conservation efforts, which often rely on translocation of individuals among populations and/or captive propagation using broodstock from healthy populations to restore depleted or extirpated populations [1,12].

Viral richness and intensity showed few differences among healthy populations. We observed the lowest levels of virus richness and intensity in the St. Croix River, where all mussels of all species were apparently healthy. Virus richness and intensity were slightly higher (but not statistically different from the St. Croix River) in apparently healthy Clinch River mucket, which we sampled from sites with active mortality of congeneric pheasantshell mussels. In the Huron River, where mucket were the only species suffering mortality, we identified approximately 4x as many viruses compared to the other populations. Several of these viruses had highly correlated read depths and co-occurred frequently, particularly in individuals collected during the mortality event. Huron River mucket had significantly higher viral richness and intensity, and the highest levels of each were associated with samples collected during the active mortality event in 2019. Subsequent analysis of mucket from the Huron River in 2020, when there was no active mortality observed in any mussel species, yielded virus richness and intensity values no different statistically from values derived from St. Croix River and Clinch River mucket. We found that viromes of apparently healthy mucket collected during the Huron River mortality event were more similar to viromes of moribund individuals collected contemporaneously than to apparently healthy mucket collected from unaffected areas. Moreover, viromes of apparently healthy mussels differed little among populations in richness and intensity.

Our observation of a novel densovirus statistically associated with cases from the Huron River is interesting as our previous studies found an association between recurring pheasantshell mortality in the Clinch River (Virginia and Tennessee, USA) and Clinch densovirus 1 [13]. However, three DNA viruses with circular genomes from the Huron River showed the most significantly elevated intensity in case compared to control mussels. This finding was not expected, given that small DNA viruses with circular genomes are ubiquitous and generally not associated with disease [33]. The strong correlations between these and the dicistroviruses suggest a pattern of generalized virome enrichment within diseased individuals. The nature of this association warrants further investigation, as does the similar association between mortality events and multiple dicistroviruses in the Huron River. We note that many of the correlations between coinfecting viruses were very high (*r* > 0.95). We suspect these values were inflated due to the parameters chosen for bioinformatic analyses, rounding of numbers, and conserved sequence regions among closely related viruses. More detailed analyses may reduce such biases.

Our results are remarkably similar to those from an investigation of viromes of healthy and unhealthy bees during periods of unexplained annual mortality, which found only 3 viral contigs in healthy bees, compared to 30 viral contigs in unhealthy bees [32]. This pattern was driven most strongly by novel members of the *Parvoviridae, Circoviridae*, and *Dicistroviridae*. Increased viral richness and intensity observed during invertebrate mortality events may reflect direct effects of one or a few viruses and secondary increased replication of other, apathogenic members of the virome due to reduced antiviral suppression by the host. These patterns could also reflect interactions among viruses, external stressors, and mussel health. Previous studies have observed increased taxonomic variation and virus proliferation in response to thermal stress in sponges [34], hydra [35], and corals [36]. Additionally, complex interactions among pathogens, environmental factors, and host attributes are suspected to drive mortality events in marine bivalves [37,38]. To evaluate the potential role of environmental stressors, we recorded basic water quality parameters (e.g., pH, dissolved oxygen, temperature, and conductivity) during sample collections and collected publicly available information on stream flow, air temperature, and precipitation for the months preceding sample collections. While none of these parameters was abnormal enough to warrant suspicion as a direct cause of the observed mortality, it is possible that such environmental factors still play an important role in shaping viromes of freshwater mussels. Future studies that simulate environmental stressors in laboratory settings may be able to more effectively examine these relationships.

Overall, our results show that unionid viromes are largely population specific and that certain viruses–and the virome as a whole–differ markedly between apparently healthy populations and populations experiencing mass mortality. Our results also show that viromes from apparently healthy and moribund mussels within populations experiencing mass mortality do not differ substantially. We suspect that longitudinal monitoring of apparently healthy individuals at these affected sites would show them to be in a pre-clinical stage of infection/disease while appearing outwardly healthy. To resolve whether the viruses identified herein are causes or consequences of disease would benefit from additional studies of mortality events in the wild as well as controlled experimental infections. Unfortunately, few tools for isolating viruses of freshwater bivalves are available, and no unionid-derived cell lines currently exist.

## Figures and Tables

**Figure 1 viruses-14-02603-f001:**
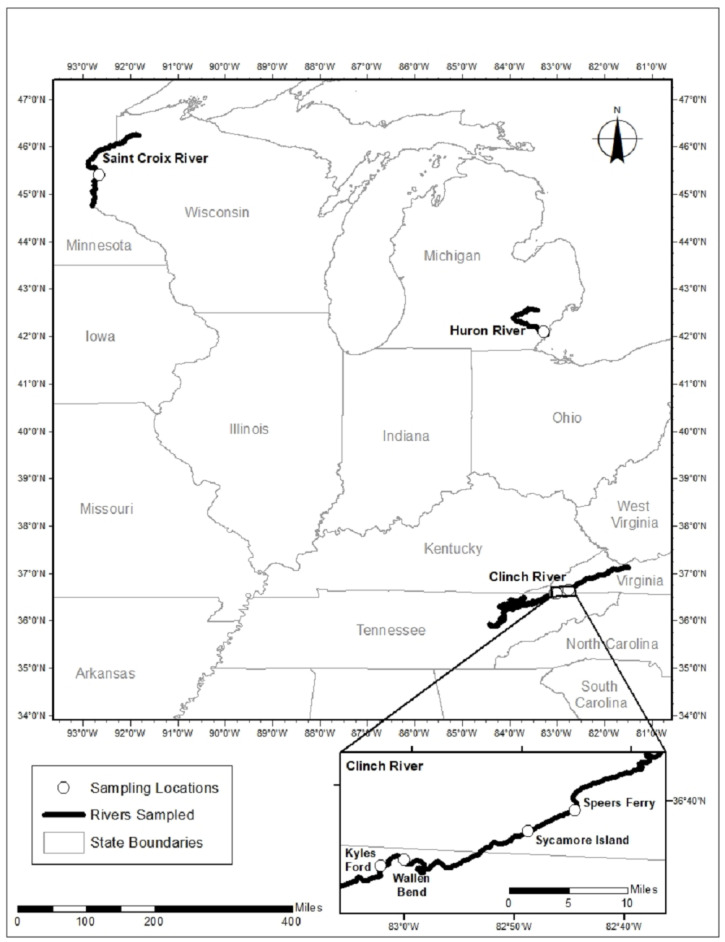
Map of sampling locations in the St. Croix River, Wisconsin, the Huron River, Michigan, and the Clinch River, Virginia and Tennessee.

**Figure 2 viruses-14-02603-f002:**
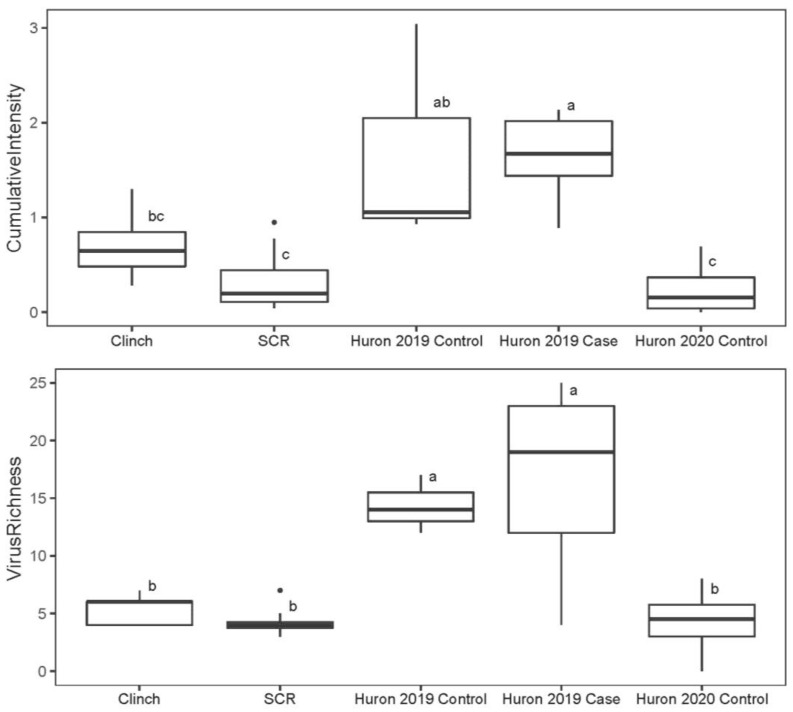
Boxplots of cumulative viral intensity (**top**) and viral richness (**bottom**) for mucket populations from the Clinch River, St. Croix River (SCR), and subgroups from the Huron River. Individual points represent outliers in the data. Boxes depict median and first and third quartiles for each group. Letters depict results from Tukey HSD analysis. Boxes with the same letters are not statistically different (i.e., adjusted *p*-value ≥ 0.05).

**Figure 3 viruses-14-02603-f003:**
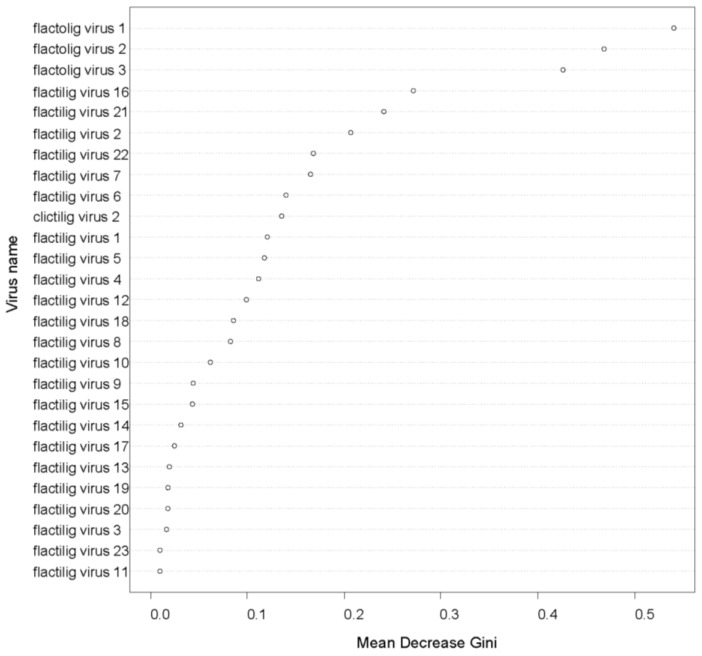
Mean decrease in Gini for each virus included in the random forest analysis of cases and controls from the Huron River. The three viruses with circular DNA genomes from the Huron River system (flactolig viruses 1, 2 and 3) had substantially higher importance values than all other viruses, followed by a picorna-like virus (flactilig virus 16) and a densovirus (flactilig virus 21) (see Appendix A for a complete description of all viruses).

**Figure 4 viruses-14-02603-f004:**
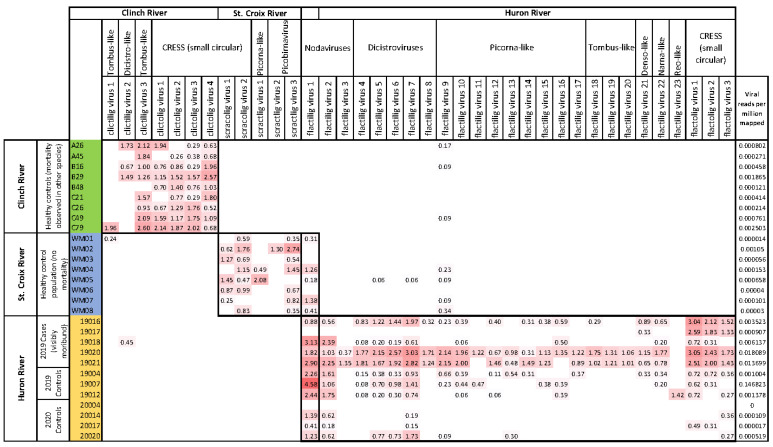
Heatmap of viral read depths (vRPM/kb) in mucket from the Clinch River, St. Croix River, and Huron River. Individual mussels are on rows, viruses are on columns. Labels identifying each virus correspond to those in Appendix A. Thick borders depict the individuals corresponding to populations from which viruses were detected. The overlapping thick border for column 13 corresponds to the only virus that was independently detected in two study populations. Viral reads per million mapped (far right column) represents the number of reads per million from the 1 million read subsample that mapped to the viruses shown in the figure. A larger version of this figure is available in Appendix A.

**Table 1 viruses-14-02603-t001:** Overview of mucket (*Actinonaias pectorosa*; syn: *Ortmanniana ligamentina*) samples used in the analyses. Full sample details are available in Appendix A.

Sample ID	Status	Date	River	Site
A26	Control	16 August 2018	Clinch	WB
A45	Control	16 August 2018	Clinch	KF
B16	Control	25 September 2018	Clinch	SF
B29	Control	25 September 2018	Clinch	SI
B48	Control	25 September 2018	Clinch	WB
C21	Control	24 October 2018	Clinch	SF
C26	Control	24 October 2018	Clinch	SF
C49	Control	25 October 2018	Clinch	SI
C79	Control	25 October 2018	Clinch	KF
WM01	Control	21 August 2018	St. Croix	Interstate
WM02	Control	21 August 2018	St. Croix	Interstate
WM03	Control	21 August 2018	St. Croix	Interstate
WM04	Control	21 August 2018	St. Croix	Interstate
WM05	Control	21 August 2018	St. Croix	Interstate
WM06	Control	21 August 2018	St. Croix	Interstate
WM07	Control	21 August 2018	St. Croix	Interstate
WM08	Control	21 August 2018	St. Croix	Interstate
19016	Case	11 September 2019	Huron	Flatrock
19017	Case	11 September 2019	Huron	Flatrock
19018	Case	11 September 2019	Huron	Flatrock
19020	Case	11 September 2019	Huron	Flatrock
19021	Case	11 September 2019	Huron	Flatrock
19004	Control	11 September 2019	Huron	Flatrock
19007	Control	11 September 2019	Huron	Flatrock
19012	Control	11 September 2019	Huron	Flatrock
20004	Control	23 September 2020	Huron	Flatrock
20014	Control	23 September 2020	Huron	Flatrock
20017	Control	23 September 2020	Huron	Flatrock
20020	Control	23 September 2020	Huron	Flatrock

## Data Availability

The sequence reads generated in this study are available at the NCBI Sequence Read Archive (SRA) database under BioProject accession PRJNA875072. Viral sequences described in this study have been deposited in GenBank under accession numbers OP441722-OP441759. Details for all individual mussel samples, viruses described in this study, viral richness, viral read depths, and statistical tests are provided in Appendix A.

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
