# Peer review of "Freshwater Mussels Show Elevated Viral Richness and Intensity during a Mortality Event"

_viruses, 2022, doi:10.3390/v14122603_

Round 1

Reviewer 1 Report (Previous Reviewer 2)

The revised manuscript is significantly improved and responsive to the reviews. I look forward to seeing this in the literature.

Reviewer 2 Report (Previous Reviewer 1)

I am happy with the revisions. 

This manuscript is a resubmission of an earlier submission. The following is a list of the peer review reports and author responses from that submission.

Round 1

Reviewer 1 Report

The manuscript by Richard et al. describes the viromes in the hemolymph of diseased and control freshwater mussels from three different sampling sites. Overall, 38 different viruses were detected, and viral diversity and “intensity” varied between samples, both being higher during the mortality event. Some of the identified viruses were associated with moribund animals or had a higher abundance in diseased animals. The study is interesting and well performed, but several parts of the manuscript are unclear or confusing and some analyses are a bit superficial. The manuscript is well written. Here are my comments:

1. The authors chose a very bizarre naming system for their viruses, where each one of them is named similarly independently on the viral family it belongs to. Additionally, for some of the viruses, they use three different ways of referring to them, two of which have progressive numbers that are different for the same virus. This got really confusing very fast. I think the authors should pick one of the three systems and stick to it. Additionally, supplementary table S2 should be moved to the main text so that the reader can always quickly go back to it to check which type of virus a paragraph is referring to. It would also be handy to have some sort of virus description (e.g., family) in figures 2 and 3 rather than just numbers.

2. Methods. The authors used rarefaction to compare read abundances between samples, but it is generally advisable to use different methods for count normalization as rarefying data may produce false positives (e.g., 10.1371/journal.pcbi.1003531). The authors should justify this choice.

3. Authors use the term “viral load” referring to read count. As the term “viral load” is generally used to describe virus genome copies or viral replicating particles (which is very different from read count) I strongly advise to use a different term (e.g., abundance) in these circumstances because this generates confusion.

4. Several times in the text you mention that you identified “novel viruses” or that viruses were “discovered” (e.g., line 179, 244 or 301: “All viruses were previously uncharacterized”) but some of them share a pretty high identity to reference strains (as per your Table S2) so they cannot be novel and they were “detected” and not “discovered”. This should be rectified.

5. Did you exclude phages from these analyses or you simply found none (except picobirnaviruses)?  

6. I noticed that you identified both cruciviruses and tombus-like viruses. Since cruciviruses are circular ssDNA viruses that acquired the gene for the cap protein from members of the Tombusviridae, I am wondering if you checked whether the tombus-like genomes you identified (at least short partial sequences) contained also the rdrp gene and verified that they are indeed tombusviruses and not cruciviruses.

7. Phylogenetic analyses as included now are poorly informative as trees were built, in many cases, with a tiny subset of known viruses and no taxonomy info about the other viruses is provided. These trees should depict how far away the identified viruses are from known taxa and more or less in which part of the tree they are located, but this is not the case for most trees presented here, which provide even fewer data than Table S2. Maybe you can at least characterize a bit better those 5 viruses that seem to be “upregulated” during the mortality event.

8. Have you checked whether the abundance of specific taxa was statistically correlated (e.g., if a specific set of taxa were consistently either all at high or low abundance, as it looks from Figure 3)?

9. Was a sampling or research permit required for this study? This should be added to the paper.

Minor:

- Lines 47 and 82. I believe the word “unionids” is used here as common name for this animal group and should, therefore, not be capitalized.

- The paragraph about sample collection is a bit confusing in the sense that the information on how many samples from where were collected is not readily available. Maybe this information can be schematically included in Figure 1?

- All figures and supplementary tables and figures have to be referenced and their order should reflect the order in which they are referenced in the text. Please, double-check this.

- Line 133. You mention here that you used the polymerase genes to make the trees but some of the viruses you identified do not have a polymerase. Please, be more specific.

- Line 230. There is a typo here “in from”.

- Lines 228-238. You say that “The three viruses with circular DNA genomes were the only viruses with significantly higher viral loads in cases when compared to controls in from 2019 and 2020.” But then below you mention other viruses that were more abundant in cases compared to controls. Is this referring to all controls pooled together? Please specify as it is confusing.

- Line 248. I believe you are referring to fig 4 here.

- Supplementary material. Official taxonomic designations (family, genus..) should be written in italics. Also, Densovirinae is not a genus but a sub-family.

- Line 306. “Picornavirales” should be written in italics.

336-7. “Clinch densovirus 1” is not a species name and it should not be written in italics. 

Reviewer 2 Report

Review for Viruses freshwater mussel viromes

The paper is generally well written.

It is succinct, providing a compelling but simple introduction, and a discussion that is to the point, embeds the work in the prior and relevant studies, and makes reasonable and thoughtful claims about the significance of the findings without over-reaching.

The paper rightly highlights the importance of elucidating the viromes of critically important species like Unionids, which are subjects of intense culture and restoration programs. All populations of animals, especially those like unionids that live in separate river drainages, are likely to have distinct viromes. Most viruses are probably not harmful in their co-evolved hosts under stable conditions. The potential to spread pathogenic strains to new host genotypes is real.

The terminology of flactilig and clictilig viruses are very confusing. Where did these terms come from and what do they denote?  This confusion exists in supplementary figure 2 as well. Please make sure this is all clear in the revised manuscript.

How many of the sequence reads were non-virus? This number, and how that relates to the virus-like preparation (differential centrifugation) and sequencing methods are helpful for other researchers to know about as the larger community investigates the viromes of other species.

Table S1 is actually a nice summary of the data that are described in the text of the results. I would recommend that this table be included in the primary manuscript, not as supplement. Authors may want to highlight the possibility that the virus diversity and prevalence in Huron in Sept 2019 may have been environmental. Or that it can’t be ruled out. In fact, what should be included in the paper to make it complete is the environmental data for each river in the 30-90 days preceding the collections. How do these values compare with other years in each location. Are they conceivably stressful, or unusually extreme in huron when the mortality occurred?

The figure with phylograms: why are there two different pages with picornaviruses? Was it the different scale (0.2 vs 0.3)?

Figure 3 shading is too faint to see the occurrence of virus 21 (and others) across all 3 geographic areas. The supplementary figure with the values added to the shaded boxes was easier to interpret.

I am not qualified to evaluate the appropriateness of the statistics in this paper. Using Gini values is novel to me